# Design and experiments of an automatic depth-adjusting double screw trencher and fertiliserning

**Caixue Zhan[1], Wenqin Ding[1], Yu Han[1], Qinghai Jiang[1], Ying Zhao[1], Liang Zhao[2], Zhiyu Song⊙[1]***

**1** Nanjing Institute of Agricultural Mechanization, Ministry of Agriculture and Rural Affairs, Nanjing, Jiangsu, China, **2** College of Mechatronics Engineering, Nanjing Forestry University, Nanjing, Jiangsu, China

* songzhiyu@caas.cn

**Data Availability Statement:** All relevant data are within the paper.

**Funding:** This research was funded by China Agriculture Research System of MOF and MARA (Grant NO. CARS-19),The Agricultural Science and

## Abstract

In response to the problems of low fertilizer application efficiency, poor operation quality, and uneven application of fertilizer by domestic tea garden trenching and fertilizing machines, an automatic depth-adjusting double screw trenching and fertilizing machine was designed. The machine uses a double spiral furrowing and fertilizer application method, which can complete the integrated operation of furrowing, fertilizer application, and mulching at one time. The key components of the machine such as the screw-type fertilizer discharger, trenching, and fertilizer application mechanism are designed using theoretical analysis, and the trenching depth is automatically adjusted through the hydraulic control system to maintain a consistent depth. A single-factor test and a quadratic regression rotary orthogonal test were conducted to select the diameter of the spiral fertilizer discharger, the rotational speed of the spiral fertilizer discharger, and the rotational speed of the trenching and fertilizer application mechanisms. Based on these tests, the fertilizer application performance of the fertilizer machine was evaluated, and a mathematical model of the fertilizer application volume and coefficient of variation was established. The influence of the test factors on the coefficient of variation was also analyzed. In the study, 58.36 and 480.35 r/min were found to be the optimal rotational speeds for spiral fertilizer discharge and trenching and fertilizer application, respectively, while 88.90 mm was found to be the optimal diameter for spiral fertilizer discharge. The coefficient of variation for the spiral fertilizer discharge was 3.95%, which meets the tea plantation's fertilizer application requirements.

## Introduction

Fertilizer application is one of the most important components of the tea production process. The quality of fertilizer application directly affects the yield and quality of tea production, and thus the appropriate fertilizer application method ensures high quality and high yield of tea [1, 2]. At present, most of the tea gardens in China use trenching machines to open a rectangular ditch followed by the application of fertilizer manually and then carrying out mulching

Technology Innovation Project of the Chinese (Grant No. CAAS-ASTIP-2022-NIAM),Yunnan Province Key R&D Special Project(Grant No. 202102AE090038),Institute-level basic scientific research business expenses project of the Chinese Academy of Agricultural Sciences (Grant No. S202103-02,Grant No.S202206),Chengdu Agricultural Science and Technology Centre Local Financial Special Funds Project "Research and Development and Demonstration of Mechanized Operation Equipment for Hilly Orchards"(Grant No. NASC2020AR03).

**Competing interests:** The authors have declared that no competing interests exist.

operations [3], which has poor fertilizer uniformity, high labour cost, significant fertilizer loss, and pollution of tea gardens [4–7]. The main types of open furrow fertilizer applicators are chain furrow openers, rotary furrow openers, and spar ploughs [8–12]. The domestic tea garden furrowing and fertilizing machines are not developed much, and thus lacking in specific tea garden operations. However, these machines are still used as fertilizing devices for fertilizing operations in tea gardens [13]. Hu Yongguang et al. [14] developed a centrifugal fertilizer spreader for tea gardens, which can achieve better fertilizer spreading uniformity through offset fertilizer spreading centrifugal discs. Dai Youhua et al. [15] developed a special type of tea garden deep pine fertilizer machine, which can be used with tractors to complete four deep pine trenching, fertilizer, and mulching operations at one time. Xiao Hongru et al. [16] developed a spiral fertilizer spreader for tea gardens, which completes the fertilizer discharge operation by means of an outer grooved wheel type fertilizer discharger with good uniformity of fertilizer discharge. Wang Linjun et al. [17] invented a hydraulic tea garden self-propelled fertilizer machine, which improved the efficiency of fertilizer application using multiple hanging mechanisms in the fertilizer box. Wang Yang developed a centrifugal fertilizer spreader and optimized the operating parameters of the spreader such as speed, blade inclination angle, blade deflection angle, and feeding amount. Li Guanghui et al. developed a cotton fertilizer machine, which mainly consists of a transmission system, a furrow opening system, a travel system, a fertilizer application system, and a mulching device. The forward speed, groove wheel opening, and grain size of fertilizer were optimized as independent variables by the response surface method by observing broken strip rate and fertilizer application accuracy.

It can be summarized that very few domestic machines are available for fertilizer application specifically for tea plantations. Moreover, the available devices can perform a specific operation at a time such as trenching, fertilizer application, and mulching, and there is no provision for automatic depth adjustment, which leads to poor efficiency of fertilizer application, poor operation quality, and uneven fertilization [18–24]. Globally, more advanced automated machinery and equipment have been developed specifically for fertilizer application in tea gardens. However, these are expensive, inflexible, unable to perform multitasking, and inconsistent with domestic tea garden planting standards. Therefore, this study proposes a novel design of an automatic depth-adjusting double-spiral trenching and fertilizer application machine, which can perform trenching, fertilizer application, and mulching operations, prevent soil accumulation, and improve fertilizer application quality and efficiency in tea gardens.

## Materials and methods

### Machine structure and working principle

**Machine structure.** In order to improve the operation efficiency, the structural layout of a double-spiral open furrow fertilizer machine suitable for the agronomic requirements of tea plantations is shown in Fig 1. It consists of a spiral fertilizer discharger, first transmission mechanism, double spiral fertilizer application mechanism, lifting mechanism, tractor, and other components.

**Working principle.** Before trenching and fertilizing operation, the depth of the left and right trenching and fertilizing mechanism is adjusted through the hydraulic system according to the agronomic requirements of the tea plantation and the growth of a tree. The operation starts with the transmission of tractive power to the fertilizing machine causing the trenching and fertilizing mechanism to descend and digs into the soil. The fertilizer then enters the trenching device through the screw-type fertilizer discharger and is finally discharged into the trench through the discharge openings of the trenching and fertilizer application mechanism. The two sets of screw knives of the trenching and fertilizer application mechanisms rotate in

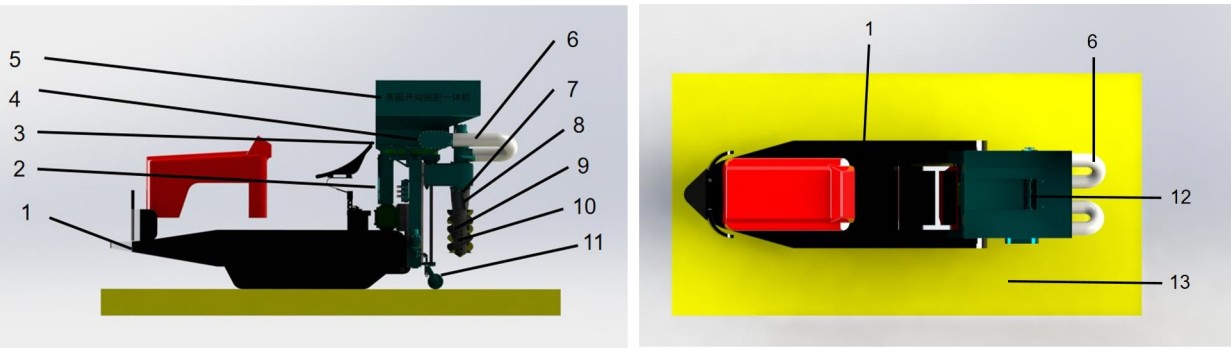

**Fig 1. Structure of the double screw trencher and fertilizer.** (a) Front view; (b) Top view. 1. tractor 2. drive shaft cover 3. fertilizer box frame 4. fertilizer discharge device 5. fertilizer box 6. fertilizer delivery pipe 7. scraper 8. trenching shaft 9. spiral blade 10. fertilizer discharge hole 11. profiled ground wheel 12. spiral fertilizer discharge device 13. tea garden.

opposite directions to achieve automatic mulching, thus realizing the integrated operation of trenching, fertilizer application, and mulching.

**Technical parameters.** To meet the agronomic requirements of tea plantations along with soil texture and fertilizer application requirements such as row spacing of 1.5–1.8 m [25] and the fertilizer application depth not less than 300 mm, the technical parameters of the double spiral open trench fertilizer applicator were determined, as shown in Table 1.

## Selection of key operating parameters of cleaning method

**Design of key components.** The double-spiral open furrow fertilizer machine has a spiral-type fertilizer discharger which comprises the fertilizer box, left fertilizer discharge port, spiral shaft, right fertilizer discharge port, left fertilizer discharge spiral, and right fertilizer discharge spiral, as shown in Fig 2. The left and right fertilizer discharge spirals are symmetrical in design.

The amount of fertilizer discharged in a rotation is an important indicator of the amount of fertilizer discharged by the spiral fertilizer discharger and depicts the performance and uniformity of fertilizer discharge [26]. Assuming one-way fertilizer discharge and axial resistance, the formula for calculating its single-ring fertilizer discharge volume (q) is as follows:

$$q = \left[ \frac{\pi(D^2 - d^2)S}{4} - bhL \right] \rho\varphi \tag{1}$$

$$L = \sqrt{\left[ \pi(D + d)/2 \right]^2 + S^2} \tag{2}$$

**Table 1. Technical parameters of the machine.**

| Items | Parameters |
|---|---|
| Structure type | Self-propelled (tracked) |
| Overall dimensions (length × width × height) (mm) | 3 260×1 425×1 555 |
| Engine type | Diesel engines |
| Maximum (fertilizer) depth/(mm) | 330 |
| Maximum trenching width/(mm) | 300 |
| Fertilizer tank volume/ m$^3$ | 0.18 |
| Operating speed/(km/h) | 0.5~1.5 |

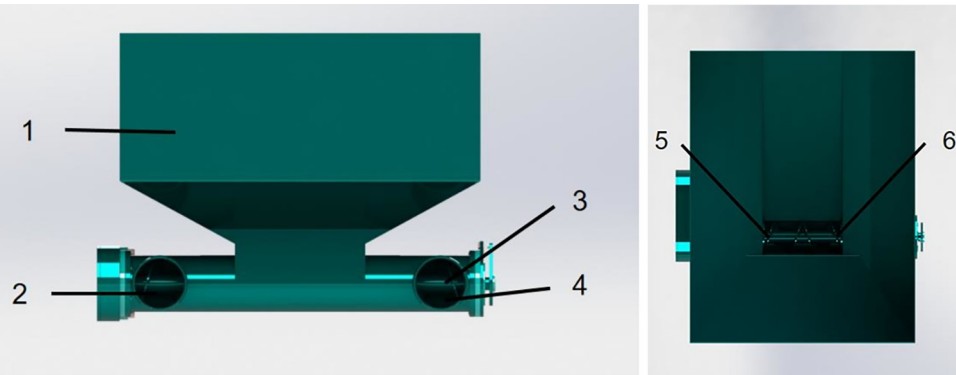

**Fig 2. Spiral fertilizer discharger structure.** (a) Front view; (b) Top view. 1. Fertilizer tank 2. left discharge port 3. spiral shaft 4. right discharge port 5. Left row fertilizer spiral 6. Right row fertilizer spiral.

$$h = (D-d)/2 \tag{3}$$

where D is the outer diameter of the spiral fertilizer drain, mm; d is the inner diameter of the spiral fertilizer drain, mm; S is the pitch of the spiral fertilizer drain, mm; b is the average thickness of the screw teeth of the spiral fertilizer drain, mm; h is the depth of the screw teeth of the spiral fertilizer drain, mm; L is the average length of the screw teeth of the spiral fertilizer drain, mm; $\rho$ is the fertilizer bulk weight, g/mm$^3$; $\varphi$ is the filling factor of the spiral fertilizer drain.

From Eq (1), it can be seen that the volume of fertilizer discharged in a single lap (q) depends on factors D, d, S, $\rho$, and $\varphi$, thus it can be varied by varying D, d, and S. The relationship between the outer diameter of the spiral fertilizer drain and the volume of fertilizer discharged is given as follows:

$$D = K\left(\frac{Q}{\varphi\lambda\varepsilon}\right)^{\frac{2}{5}} \tag{4}$$

$$K = \left(\frac{1}{47cA}\right)^{\frac{2}{5}} \tag{5}$$

where Q is the spiral fertilizer discharge volume, t/h; A is the combined material properties factor; K is the material combination factor; c is the ratio of pitch to diameter; $\lambda$ is the mass per unit volume of material, t/m3; $\varepsilon$ is the transport coefficient.

For continuous operation of the fertilizer discharge machine, the fertilizer consumption is given as follows:

$$Q_S = v_m g/s \tag{6}$$

where Qs is the volume of fertilizer supplied, t/h; g is the amount of fertilizer supplied for a distance forward, t; s is the forward distance.

Based on agronomic requirements for fertilization in tea plantations, 1.8 kg of fertilizer is required for a single tea plant, so each side supplies 0.9 kg of fertilizer. For the row spacing of trees as 1.5~1.8 m, the speed of trenching and fertilizing machine as 1000 m/h, the fertilizer application volume Qs is estimated to be 0.6 t/h, which is also verified by the expression given in Eq (6). Subsequently, the filling factor of fertilizer $\varphi$ can be obtained as 0.25, the

comprehensive material characteristic coefficient A of fertilizer as 28, the material unit volume mass of fertilizer as 1.2t/m³, the ratio of pitch to the diameter as 0.9, and the conveying coefficient as 0.9. Using these parameters in Eqs (4) and (5), the outer diameter of spiral fertilizer discharge is found to be 91 mm. As the spiral blade should be designed into a standard series, it can be determined that the outer diameter of the spiral fertilizer discharge D is 88 mm, the inner diameter of the spiral fertilizer discharge d is 32 mm, the pitch of the spiral fertilizer discharge is S is 80 mm, the average thickness of the screw teeth b is 3 mm. Substituting these parameters into Eqs (1)—(3), the single rotation of fertilizer discharge q is found to be 197.5 g.

**Trenching and fertilizing mechanism.** *Design of trenching and fertilizing mechanism.* The trenching and fertilizing mechanism mainly consists of a hydraulic motor, fertilizer discharge screw conveyor, inlet hole, shaft cover, auger blade, discharge hole, and auger tip, as shown in Fig 3. A double auger trenching and fertilizing mechanism is used for trenching and fertilizing. The augers of the trenching and fertilizing mechanism are symmetrical left and right and rotate at opposite speeds. This trenching and fertilizing mechanism is powered by a hydraulic motor and has three auger sections.

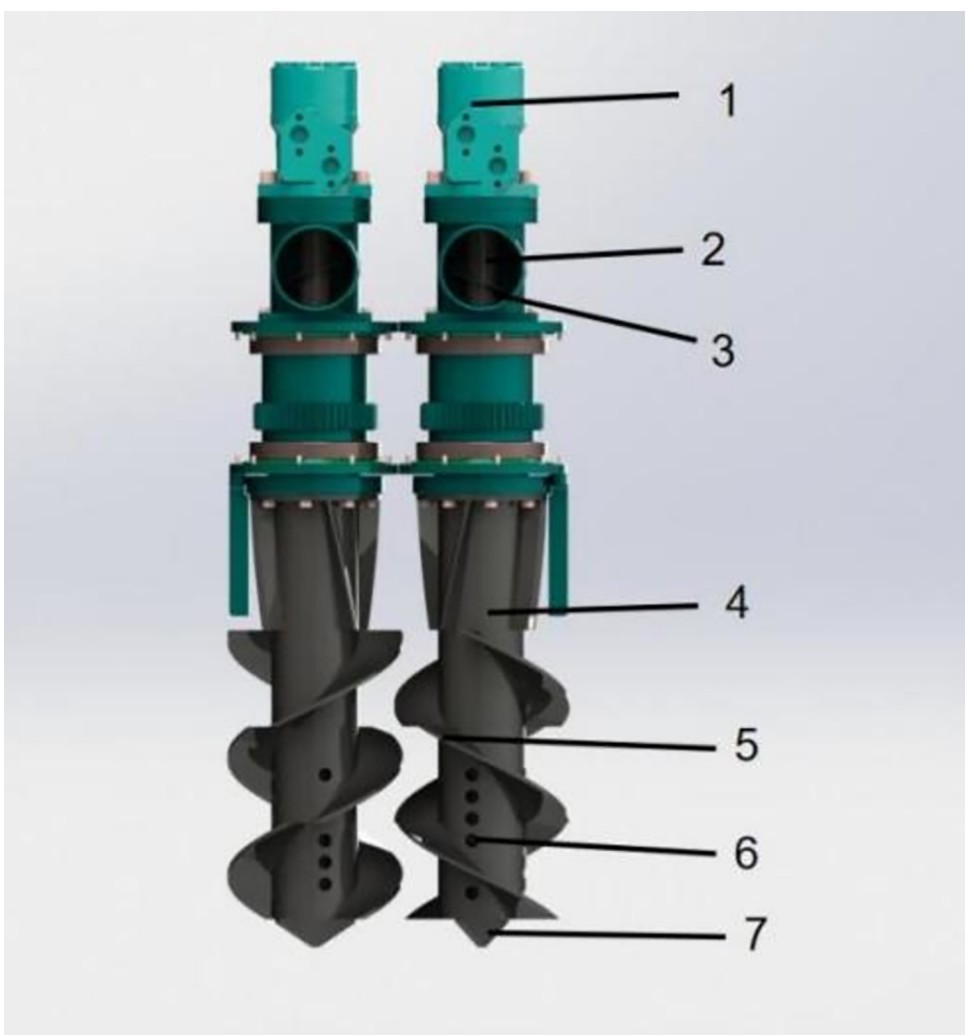

**Fig 3. Structure of the trenching and fertilizing mechanism.** 1. hydraulic motor 2. fertilizer discharge screw conveyor 3. fertilizer inlet hole 4. shaft cover 5. auger blade 6. fertilizer discharge hole 7. auger tip.

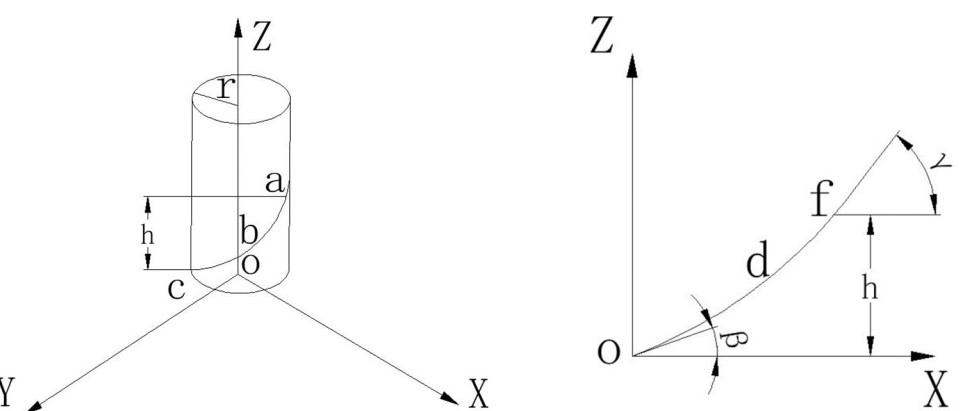

**Fig 4.** Diagram of spiral blades (a) Spiral line diorama; (b) Spiral expansion diagram. Among them, r is the radius of the trenching knife; h is the height of point f; β is the helix angle at c; γ is the helix angle at f.

The spiral blade is a key component of the trenching and fertilizer application mechanism and therefore its parameters directly influence the quality of the trenching and the power consumption of the double-spiral trenching and fertilizer application machine. During the operation of the trenching and fertilizer application mechanism, the spiral blade mills the ground and the milled soil rises along the spiral blade and is transferred to the trench during the reverse operation. The spiral blades of the trenching and fertilizing mechanism have variable pitch and are made up of numerous tangential lines, as shown in Fig 4.

The spiral blades of variable pitch are mounted on the cylinder that rotates at a uniform speed along the Z-axis. The generating line is the locus of point b which is formed by uniformly accelerating the cylinder along the Z-axis, and its path trajectory can be conceived as a variable pitch spiral line. Expanding it on the plane ZOX gives a parabola (as shown in Fig 4b), while the angle of the helix at the starting point of the helix c is $2\pi n_1$. The equation of the helix can be determined from r, β, $n_1$, and h when point b reaches the position of point a when it rises by height h along the Z axis.

$$\begin{cases} x = r\cos 2\pi rn \\ y = r\sin 2\pi rn \\ z = a(2\pi r)^2 n^2 + (2\pi r\tan\beta)n \end{cases} \quad (7)$$

$$a = \frac{h - 2\pi rn_1\tan\beta}{(2\pi rn_1)^2} \quad (8)$$

where r is the radius of the trenching knife, mm; n is the number of helix turns; a is the angle of the helix; $n_1$ is the number of helix turns at b; β is the angle of the helix at c, °; h is the height of point f, mm.

The trenching and fertilizing mechanism mills out the soil by the spiral blade operation. Keeping the agronomic requirements for fertilizing tea gardens into consideration, the trenching width and depth were taken as 200–300 mm and 300–400 mm, respectively. The trenching and fertilizing mechanism is designed by computing the key design parameters from Eqs (7) and (8), and are shown in Table 2.

*Design of the hydraulic system.* The hydraulic control system acts as a control key for the double screw trencher and fertilizer mechanism, which consists of a hydraulic pump, profiling valve, profiling ground wheel, first manual reversing valve, second manual reversing valve,

Table 2. Key design parameters of the trenching and fertilizing mechanism.

| Items | Parameters |
|---|---|
| Inner diameter of spindle/(mm) | 88 |
| Outer diameter of spindle/(mm) | 100 |
| Height of spindle/(mm) | 478 |
| Spiral blade thickness/(mm) | 5.5 |
| Initial helix angle / (°) | 30 |
| Maximum rotation speed/(rad/s) | 4 |

speed control valve, and balance valve set, as shown in Fig 5. The profiling wheel is mounted on a profiling valve, the profiling valve ports A and B are connected to the rod and rodless chambers of the first hydraulic cylinder via a balancing valve set. Port P of the profiling valve is connected to port B of the first manual reversing valve, and port A of the first manual reversing valve is connected to the rod chamber of the first hydraulic cylinder via a balancing valve set. Ports A and B of the second manual reversing valve are connected to the rod and rodless chambers of the second hydraulic cylinder and ports P of the first and second manual reversing valves are connected to the hydraulic pump.

The height of the trenching and fertilizing mechanism is controlled by the first hydraulic cylinder. Initially, the trenching and fertilizing mechanism is inserted into the soil surface while the profiling wheel is in contact with the ground. As the double spiral trencher and fertilizer mechanism moves forward, the profiling wheel undulates with the bottom surface, driving the control lever of the corresponding profiling control valve up or down. The profiling valve thus connects with the rod or rodless chamber of the first hydraulic cylinder, which drives the corresponding trenching and fertilizing mechanism until the control lever of the profiling control valve returns to its initial position.

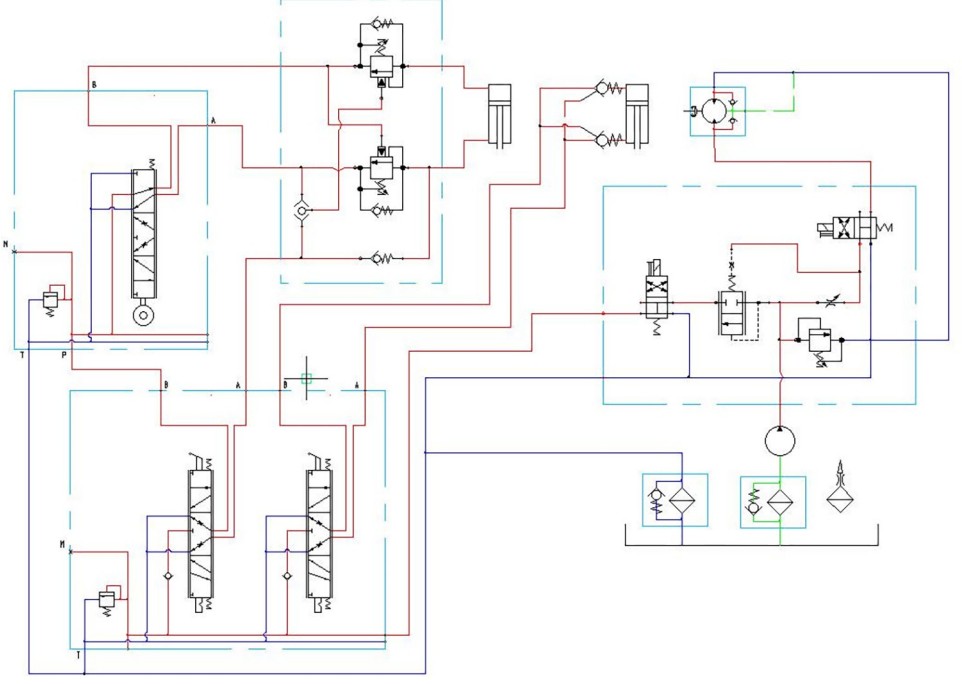

Fig 5. Hydraulic system schematics.

When the fertilizer spreader moves forward, the profiling wheel is always in contact with the working area. If the terrain level is raised, the profiling wheel moves up and triggers the control lever of the profiling valve, which connects the profiling valve with the rod chamber of the first hydraulic cylinder. The hydraulic oil from the hydraulic pump enters the rodless chamber of the first hydraulic cylinder, causing the telescopic rod of the first hydraulic cylinder to move downwards, which in turn drives the spiral trenching mechanism and the material transfer mechanism downwards. The hydraulic system is triggered by the profiling valve and the profiling wheel to ensure that the spiral trenching mechanism and the material transfer mechanism rise and fall in sync with the terrain, thus making the trenching depth consistent.

## Test conditions and methods

**Machine processing and test conditions.** The fertilizing machine was machined and assembled as shown in Fig 6. In order to optimize the structure and performance parameters of the double-spiral trenching and fertilizer machine, tests were carried out in June 2021 in a standard tea garden with a row spacing of 150 cm and a plant spacing of 30 cm located at Shenhang Forest, Tea and Fruit Seed Technology Co. The Linkwo compound fertilizer produced by Anhui Linkwo Biotechnology Co fertilizer was used during the tests. The fertilizer had a moisture content of 1.18% and the average diameter of the fertilizer particles was 4.1 mm.

**Test methods.** A one-way test with quadratic regression rotated orthogonal test involving the coefficients of variation of fertilizer discharge and stability of fertilizer discharge was conducted to examine fertilizer application performance of a double-screw trenching and fertilizing machine. The influence of each factor on the test indices and the optimal range of parameters were determined through a single-factor test using the recommended technical specification of quality evaluation for fertilizing machinery (NY/T 1003–2006).

## Results and analysis

### One-factor tests and analysis

After several pre-experiments, the key parameters such as the diameter of the spiral discharger, the rotational speed of the spiral discharger, and the rotational speed of the trenching and fertilizer application mechanism were selected as the test factors for the single-factor test, and the test results are shown in Fig 7.

The coefficient of variation decreases with the increase in speed of the spiral fertilizer discharge up to 60 r/min, at which it attains a minimum value of 3%, beyond which it increases. The fertilizer discharge increases with the increase in the speed of fertilizer discharge.

The coefficient of variation tends to decrease as the diameter of the spiral fertilizer discharger increases and reaches a minimum value of 4.1% at 90 mm fertilizer discharge diameter, while the amount of fertilizer discharge gradually increases with the increase in diameter of fertilizer discharge.

The coefficient of variation decreases as the rotational speed of the trenching and fertilizer application mechanism increases and attains a minimum value of 3.5% at 480 r/min, however, the fertilizer discharge tends to rise gradually.

### Quadratic-regression rotatable orthogonal experiment and analysis

In order to determine the optimal range of operating parameters in accordance with the results of the single-factor test, the speed of the spiral fertilizer discharger was determined to be from 50 to 70 r/min, the diameter of the spiral fertilizer discharger to be from 86 to 94 mm, and the speed of the trenching and fertilizer application mechanism to be from 470 to 490 r/min. The

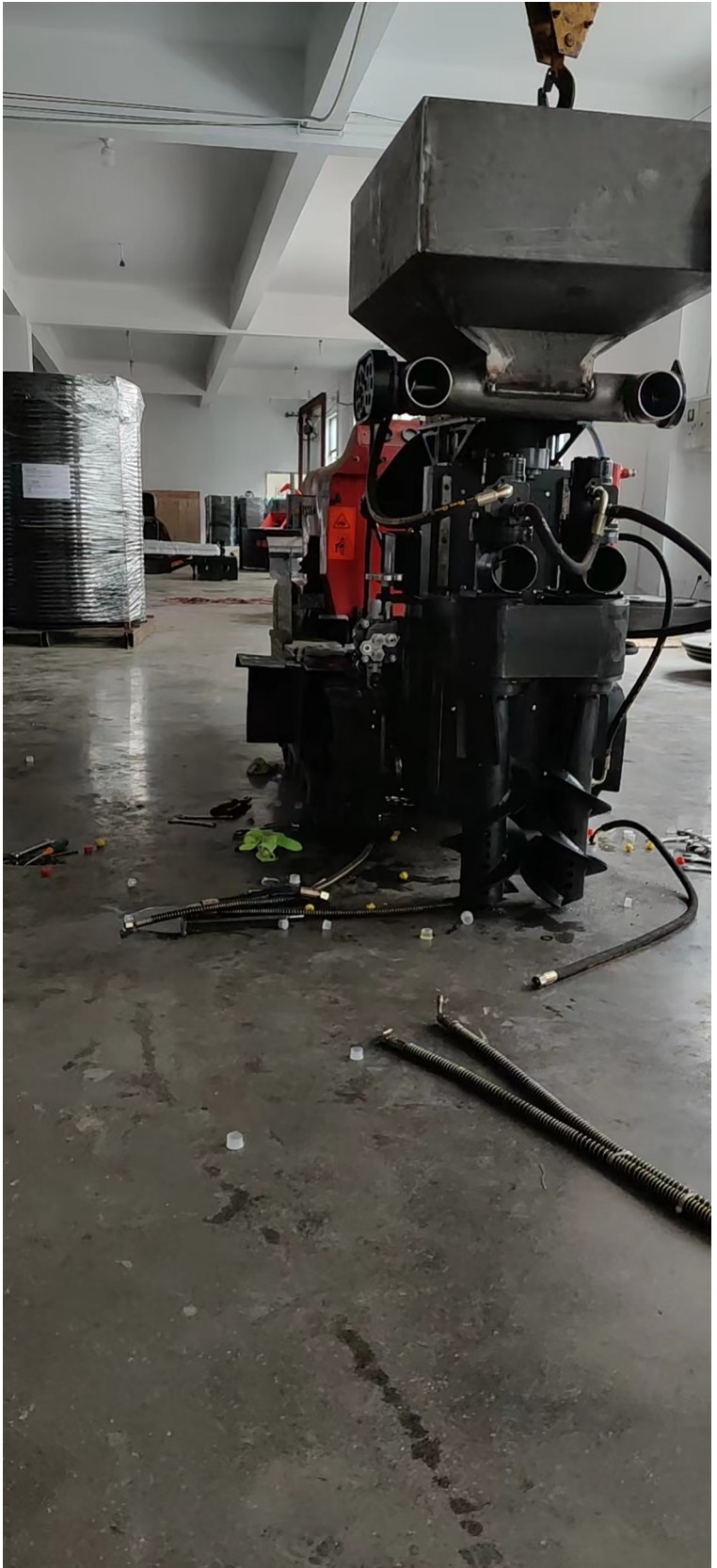

**Fig 6. Machining and assembly of fertilizing machine.**

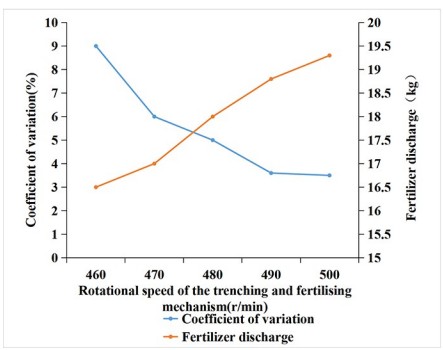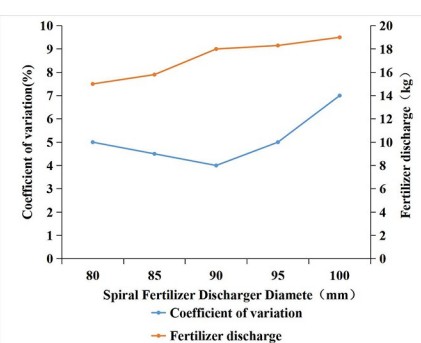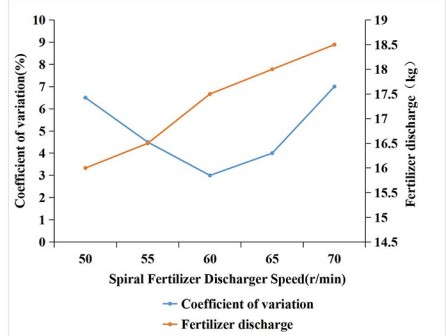

**Fig 7. Single-factor test.**

test factor level coding is shown in Table 3. The experimental design scheme and results are shown in Table 4.

**Fertilizer discharge Y1.** ANOVA was performed on the test data shown in Table 4 using Design Expert software, yielding the regression data shown in Table 5. According to Table 5, the fit of the fertilizer discharge model $Y_1$ is highly significant (P < 0.01). The failure to fit term, P = 0.5976, is not significant, indicating that there are no other factors affecting the indicator. The ascending order of factors based on their influence on the fertilizer discharge is the diameter of the spiral discharger $X_2$, the speed of the spiral discharger $X_1$, and the speed of the trenching and fertilizer application mechanism $X_3$. The quadratic term ($X_3^2$) had a significant effect, and after excluding the insignificant factors in the interaction term, the regression model is expressed as:

$$Y_1 = 18.11 + 0.68X_1 + 0.91X_2 - 0.59X_3 - 0.5X_3^2 \tag{9}$$

**Coefficient of variation Y2.** ANOVA was also performed on the test data in Table 4 using Design Expert software to see the fit of the coefficient of variation model $Y_2$ and found it highly significant (P < 0.01) as shown in Table 5. The failure to fit term, P = 0.0665, was not significant, thus indicating no other influencing factor. The descending order of factors based on their influence on the fertilizer discharge is the diameter of the spiral discharger $X_2$, the speed of the spiral discharger $X_1$, and the speed of the trenching and fertilizer application mechanism $X_3$. The interaction term ($X_1X_2$) and the quadratic term ($X_1^2$, $X_3^2$) between the diameter of the spiral fertilizer discharger and the rotational speed of the spiral fertilizer discharger are significant, and after excluding the insignificant factors in the interaction term, the regression model is expressed as:

$$Y_2 = 4.67 + 0.72X_1 + 0.75X_2 - 0.59X_3 + 0.83X_1X_2 + 0.94X_1^2 + 0.31X_3^2 \tag{10}$$

**Table 3. Test factors and levels.**

| Level | Spiral fertilizer discharge speed $X_1$/r/min | Diameter of the spiral fertilizer drainer $X_2$/mm | Rotational speed of the trenching and fertilizing mechanism $X_3$/r/min |
|---|---|---|---|
| -1.682 | 50 | 86 | 470 |
| -1 | 55 | 88 | 475 |
| 0 | 60 | 90 | 480 |
| 1 | 65 | 92 | 485 |
| -1.682 | 70 | 94 | 490 |

**Table 4. Experimental design scheme and test results.**

| Test No. | Factor level | | | Test results | |
|:---:|:---:|:---:|:---:|:---:|:---:|
| | $X_1$ | $X_2$ | $X_3$ | $Y_1$ | $Y_2$ |
| 1 | -1 | -1 | -1 | 16.1 | 5.6 |
| 2 | 1 | -1 | -1 | 17.8 | 5.7 |
| 3 | -1 | 1 | -1 | 18.1 | 6 |
| 4 | 1 | 1 | -1 | 18.6 | 9.2 |
| 5 | -1 | -1 | 1 | 15.6 | 6.5 |
| 6 | 1 | -1 | 1 | 16.5 | 5.3 |
| 7 | -1 | 1 | 1 | 17.5 | 5.2 |
| 8 | 1 | 1 | 1 | 18.3 | 7.5 |
| 9 | -1.682 | 0 | 0 | 16.2 | 5.1 |
| 10 | 1.682 | 0 | 0 | 19.5 | 8.3 |
| 11 | 0 | -1.682 | 0 | 15.7 | 4.9 |
| 12 | 0 | 1.682 | 0 | 19.2 | 8.1 |
| 13 | 0 | 0 | -1.682 | 18.7 | 6.7 |
| 14 | 0 | 0 | 1.682 | 15.5 | 3.1 |
| 15 | 0 | 0 | 0 | 18.5 | 4.1 |
| 16 | 0 | 0 | 0 | 18.8 | 3.2 |
| 17 | 0 | 0 | 0 | 19.6 | 4.3 |
| 18 | 0 | 0 | 0 | 18.4 | 5.1 |
| 19 | 0 | 0 | 0 | 18.2 | 3.7 |
| 20 | 0 | 0 | 0 | 18.5 | 3.9 |

**Effect of interaction of factors on the conformity index.**  Based on the analysis of experimental data, the effects of speed and diameter of the spiral fertilizer discharger, and the speed of the trenching and fertilizer application mechanism on the coefficient of variation $Y_2$ are shown using the response surface in Fig 8.

Fig 8A–8C show the effect of the interaction of the test factors on the test indicators. Fig 8A shows the response surface of the interaction between the speed and the diameter of the spiral discharger on the coefficient of variation $Y_2$ at 480 r/min for the open furrow fertilizer application mechanism. The coefficient of variation tends to decrease and then increase as the speed and diameter of the spiral fertilizer discharger increase. At relatively low rotational speed and the diameter of the spiral fertilizer, the relative movement of the fertilizer and the discharger is slower, and the internal friction of the fertilizer is greater, making fertilizer discharge at a larger coefficient of variation difficult. The coefficient of variation decreases as the rotational speed and the diameter of the spiral fertilizer discharger increase, resulting in faster relative movement of the fertilizer and the discharger and less internal friction of the fertilizer, which facilitates the easy discharge of fertilizer. The coefficient of variation increases as the rotational speed and the diameter of the spiral fertilizer discharger becomes too large, the relative movement of the fertilizer and the fertilizer discharger becomes too intense, and the fertilizer discharge efficiency decreases as a result of mutual collisions. The coefficient of variation remains low for spiral fertilizer discharge speeds of 55–65 r/min and spiral fertilizer discharge diameters of 88–92 mm.

Fig 8B shows the response surface of the interaction between the rotational speed of the spiral discharger and the rotational speed of the open furrow fertilizer application mechanism on the coefficient of variation $Y_2$ for a spiral discharger diameter of 90 mm. The plot shows that the coefficient of variation tends to decrease and then increase as the rotational speed of the spiral fertilizer discharger and the rotational speed of the trenching and fertilizer application

**Table 5. Analysis of variance table.**

| Source of variance | Fertilizer discharge $Y_1$ | | | | Coefficient of variation $Y_2$ | | | |
|---|---|---|---|---|---|---|---|---|
| | Sum of squares | Freedom | F | P | Sum of squares | Freedom | F | P |
| Model | 25.69 | 9 | 5.42 | 0.0071** | 26.40 | 9 | 39.25 | < 0.0001** |
| $X_1$ | 6.54 | 1 | 12.43 | 0.0055** | 2.07 | 1 | 14.68 | 0.001** |
| $X_2$ | 11.23 | 1 | 21.35 | 0.0010** | 2.39 | 1 | 83.14 | < 0.0001** |
| $X_3$ | 4.78 | 1 | 9.09 | 0.0130* | 0.080 | 1 | 36.38 | < 0.0001** |
| $X_1X_2$ | 0.21 | 1 | 0.40 | 0.5406 | 3.78 | 1 | 0.77 | 0.3901 |
| $X_1X_3$ | 0.031 | 1 | 0.059 | 0.8124 | 0.00125 | 1 | 2.73 | 0.1136 |
| $X_2X_3$ | 0.10 | 1 | 0.19 | 0.6703 | 0.061 | 1 | 32.13 | < 0.0001** |
| $X_1^2$ | 0.18 | 1 | 0.35 | 0.5680 | 6.01 | 1 | 19.80 | 0.0002** |
| $X_2^2$ | 0.93 | 1 | 1.77 | 0.2129 | 4.77 | 1 | 94.95 | < 0.0001** |
| $X_3^2$ | 2.06 | 1 | 3.91 | 0.0761 | 10.61 | 1 | 7.26 | 0.0136* |
| Residuals | 5.26 | 10 | | | 3.52 | 10 | | |
| lack of fit | 2.42 | 5 | 0.85 | 0.5671 | 1.49 | 5 | 1.58 | 0.2318 |
| Pure error | 2.84 | 5 | | | 2.03 | 5 | | |
| Cor total | 30.95 | 19 | | | 6102.53 | 19 | | |

Note: P<0.01 (highly significant, **), P<0.05 (significant, *).

mechanism increase. Significant internal friction is observed in the fertilizer and the rotational speed of the trenching and fertilizer application mechanism for low rotational speeds of the spiral fertilizer discharger and the trenching and fertilizer application mechanism, resulting in a slower relative movement between fertilizer and discharger, which is not favorable for the fertilizer discharge. The coefficient of variation decreases as the rotational speed of the spiral fertilizer discharger and the trenching and fertilizer application mechanism increase, resulting in faster relative movement of the fertilizer and the discharger and less internal friction of the fertilizer, which facilitates the quick and easy discharge of the fertilizer. Fertilizer tends to adhere to the discharge and open trench application mechanisms under the effect of centrifugal force when the rotational speed of the spiral fertilizer discharge mechanism and the open trench application mechanism becomes too large, reducing the effectiveness of fertilizer discharge. The coefficient of variation is found low at rotational speeds of 55 to 65 r/min for the spiral fertilizer discharge and rotational speeds of 475 to 485 r/min for the open furrow fertilizer application mechanism.

Fig 8C shows the response surface of the interaction between the diameter of the spiral discharger and the speed of the open furrow fertilizer application mechanism on the coefficient of variation Y2 at a speed of 60 r/min of the spiral discharger. It can be observed from the plot that the coefficient of variation decreases by increasing the diameter of the auger and the speed of the trenching and fertilizer application mechanism, attaining minima and then increasing. At a large coefficient of variation, the lower values of the diameter of the spiral fertilizer discharger and the speed of the trenching and fertilizer application mechanism are insufficient to achieve effective fertilizer discharge. With a larger diameter of the fertilizer, the fertilizer discharge declines even with a large diameter of the spiral fertilizer discharger. While at a higher speed of the trenching and fertilizer application mechanism, fertilizer tends to adhere to the trenching and fertilizer application mechanism under the action of centrifugal force, leading to a reduction in the efficiency of fertilizer discharge, for increasing values of coefficient of variation. For the diameter for the rotary fertilizer discharge from 88 to 92 mm and the rotational speed of trenching and fertilizer application mechanism from 475 to 485 r/min, the coefficient of variation is observed to be low.

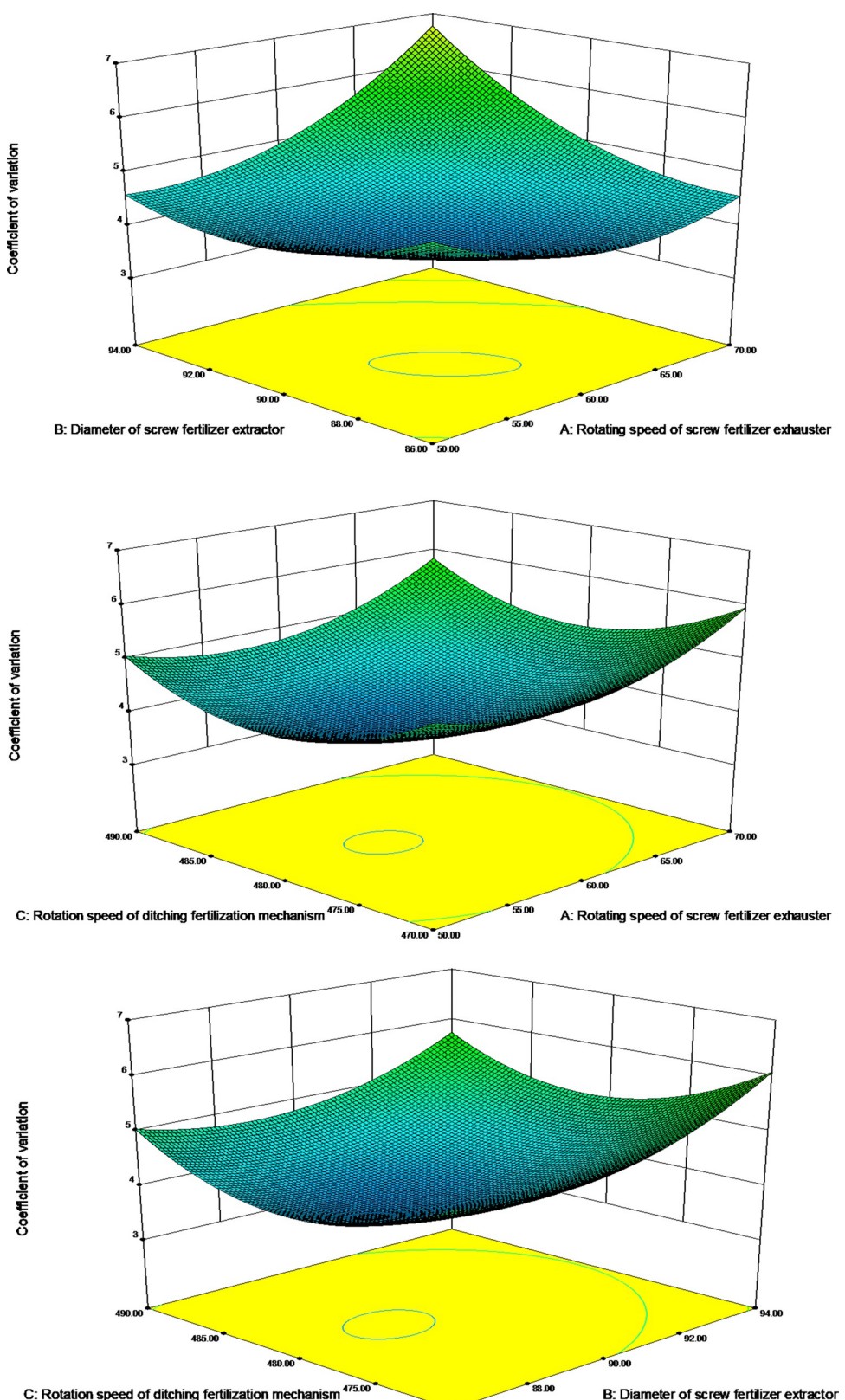

**Fig 8. Response surface for the effect of each factor on the coefficient of variation.** (a) Open trenching and fertilizing mechanism at 480 r/min; (b) Spiral fertilizer discharge at 90 mm diameter; (c) Spiral fertilizer discharge at 60 r/min.

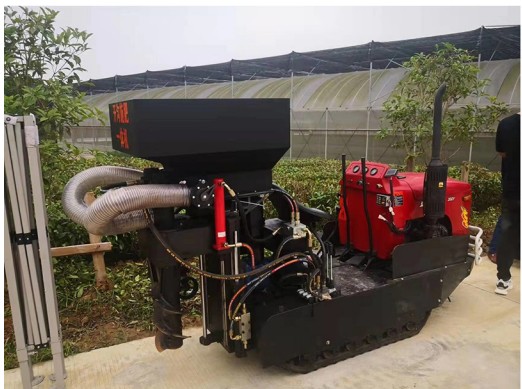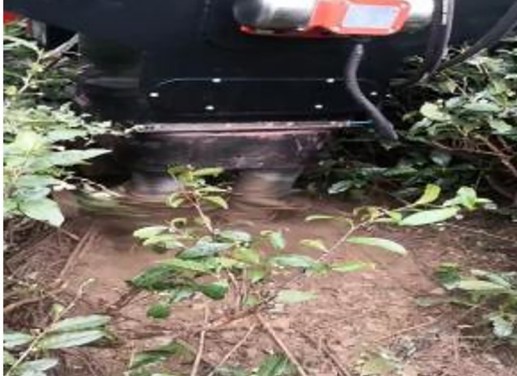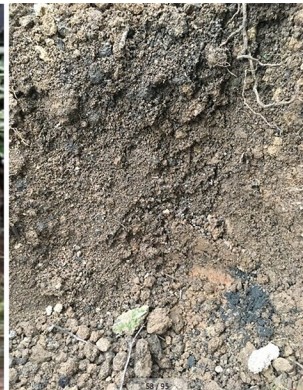

**Fig 9. Performance test of the double screw trencher and fertilizer.**

**Parametric optimization.** Based on the requirements of the technical specification of quality evaluation for fertilizing machinery (NY/T 1003–2006) for the agronomic requirements of tea plantation, the smaller values of the coefficient of variation in the fertilizer through discharge of the double spiral furrow spreader are preferred. Therefore, the relevant parameters of the spreader need to be further optimized and the corresponding coefficient of variation in the fertilizer discharge under optimized conditions needs to be obtained. In order to find the optimal solution for each relevant factor, the regression model of the coefficient of variation is used as the objective function. The range of parameters, for which it has been solved, is used as the constraint, and the minimum value of the objective function is obtained by using Design Expert software. The objective function and constraints for optimization are as follows:

$$
\begin{cases}
\min y_2(x_1 x_2 x_3) \\
\text{s.t.}
\begin{cases}
50\text{r/min} \le x_1 \le 70\text{r/min} \\
80\text{mm} \le x_2 \le 94\text{mm} \\
470\text{r/min} \le x_3 \le 490\text{r/min}
\end{cases}
\end{cases}
\tag{11}
$$

The optimum set of parameters was obtained as the rotational speed of spiral fertilizer discharge as 58.36 r/min, diameter of the spiral discharge as 88.90 mm, and speed of trenching and fertilizer application mechanism as 480.35 r/min, where the coefficient of variation was 3.95%.

The working parameters are determined based on parametric optimization of the test parameters collected in August 2021 in a trial field of the tea garden of Shenhang Forest, Tea and Fruit Technology Co Ltd in Liyang, Changzhou, Jiangsu (as shown in Fig 9). Based on an optimal set of working parameters, the coefficient of variation of fertilizer discharge was 4.52%. The validation results were more consistent with the optimization results, and the double-spiral furrow spreader design met agronomic requirements in terms of fertilizer application.

## Conclusions

1. An automatic depth-adjusting double-screw trenching and fertilizer applicator is designed, which can realize the integrated operation of trenching, fertilizer application, and

mulching. In order to achieve automatic trenching depth adjustment through the built-in hydraulic control system, the screw fertilizer discharger and trenching and fertilizer application mechanism are analyzed to achieve this goal. This improves the efficiency of trenching and fertilizer application in tea plantations, solving the difficult problem of inconsistent fertilizer application depth in tea plantations.

2. Through theoretical analysis, the screw fertilizer distributor, spiral ditching and fertilizing mechanism and control system are designed, and the ditching depth is automatically adjusted by the hydraulic control system to improve the operation efficiency and fertilizing effect of the ditching and fertilizing machine.

3. Field trials were conducted to determine the primary and secondary relationships affecting fertilizer discharge performance, using the orthogonal test. The results showed that the optimized value of the auger discharger speed was 58.36 r/min, the diameter of the auger discharger was 88.90 mm, and the trenching and fertilizer application mechanism speed was 480.35 r/min, at a coefficient of variation of 3.95%, proving the reliability of the optimization results through validation tests.

## Acknowledgments

Thanks to Professor Zhiyu Song for his guidance. Thanks to Wenqin Ding for their experimental program discussion and experimental assistance.

## Author Contributions

**Conceptualization:** Qinghai Jiang.

**Data curation:** Caixue Zhan, Wenqin Ding.

**Formal analysis:** Caixue Zhan, Wenqin Ding.

**Funding acquisition:** Zhiyu Song.

**Investigation:** Caixue Zhan, Wenqin Ding, Yu Han, Qinghai Jiang.

**Methodology:** Caixue Zhan, Wenqin Ding, Yu Han.

**Project administration:** Caixue Zhan.

**Resources:** Caixue Zhan.

**Software:** Yu Han, Ying Zhao.

**Supervision:** Yu Han, Zhiyu Song.

**Validation:** Qinghai Jiang, Liang Zhao, Zhiyu Song.

**Visualization:** Caixue Zhan, Wenqin Ding, Qinghai Jiang.

**Writing – original draft:** Caixue Zhan, Wenqin Ding.

**Writing – review & editing:** Caixue Zhan, Wenqin Ding, Qinghai Jiang.

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
