## [Decision Letter · Decision Letter 0]

1 Sep 2022

PONE-D-22-07555Design and experiments of an automatic depth-adjusting double screw trencher and fertiliserningPLOS ONE

Dear Dr. song,

Thank you for submitting your manuscript to PLOS ONE. After careful consideration, we feel that it has merit but does not fully meet PLOS ONE’s publication criteria as it currently stands. Therefore, we invite you to submit a revised version of the manuscript that addresses the points raised during the review process. Please revise.

We look forward to receiving your revised manuscript.

Kind regards,

Academic Editor

PLOS ONE

“This research was funded by China Agriculture Research System of MOF and MARA

(Grant NO. CARS-19),The Agricultural Science and Technology Innovation Project of the Chinese

(Grant No. CAAS-ASTIP-2022- NIAM),Yunnan Province Key R&D Special Project(Grant No. 202102AE090038),Research on key technologies and devices for unmanned orchards(Grant No. S202103-02),Chengdu Agricultural Science and Technology Centre Local Financial Special Funds Project "Research and Development and Demonstration of Mechanized Operation Equipment for Hilly Orchards"(Grant No. NASC2020AR03).

Acknowledgments: The authors would like to acknowledge Jiangsu Changzhou Liyang Shenhang Forest, Tea and Fruit Seed Technology Co. for providing the test site for this study and the help in the test process.”

“Jiangsu Changzhou Liyang Shenhang Forest, Tea and Fruit Seed Technology Co. for providing the test site for this study and the help in the test process.”

“Jiangsu Changzhou Liyang Shenhang Forest, Tea and Fruit Seed Technology Co. for providing the test site for this study and the help in the test process.”

6. PLOS requires an ORCID iD for the corresponding author in Editorial Manager on papers submitted after December 6th, 2016. Please ensure that you have an ORCID iD and that it is validated in Editorial Manager. To do this, go to ‘Update my Information’ (in the upper left-hand corner of the main menu), and click on the Fetch/Validate link next to the ORCID field. This will take you to the ORCID site and allow you to create a new iD or authenticate a pre-existing iD in Editorial Manager. Please see the following video for instructions on linking an ORCID iD to your Editorial Manager account: https://www.youtube.com/watch?v=_xcclfuvtxQ.

Reviewers' comments:

Reviewer's Responses to Questions

**Comments to the Author**

1. Is the manuscript technically sound, and do the data support the conclusions?

Reviewer #1: Yes

2. Has the statistical analysis been performed appropriately and rigorously? 

Reviewer #1: Yes

3. Have the authors made all data underlying the findings in their manuscript fully available?

Reviewer #1: Yes

4. Is the manuscript presented in an intelligible fashion and written in standard English?

Reviewer #1: Yes

5. Review Comments to the Author

Reviewer #1: * English editing.

* Figure 7 is not clear.

* Please state the country of the Technical specification of quality evaluation for fertilizing machinery(NY/T 1003-2006).

* Please give explanation whey the coefficient of variation shows a trend of decreasing as the speed of the spiral fertilizer

drain increases.

* In introduction , no research papers using RSM were reviewed, specially in fertilization machines.

* Provide φ (the filling factor of the spiral fertilizer drain).

* Does the fertilizer moisture content and its diameter had effect on coefficient of variation ? please give data orrefences.

6. PLOS authors have the option to publish the peer review history of their article (what does this mean?). If published, this will include your full peer review and any attached files.

Reviewer #1: No

---

## [Author Response · Author response to Decision Letter 0]

13 Oct 2022

Dear Editors and Reviewers

Thank you very much for your email regarding my manuscript (PONE-D-22-07555). We really thank you for giving us a valuable opportunity to improve our manuscript. According to your suggestion and referees’ comments, we have carefully revised the manuscript and addressed all the comments.

The revised version with changes in RED has been submitted electronically via the Web. Enclosed please find a letter detailing our revisions and reply to the comments. I hope the revised version is suitable for publication.

I am looking forward to hearing from you.

Best regards and wishes,

Zhiyu Song

To Reviewer #1

Comment 1: English editing.

Response: Thank you for your valuable advice.English editing has been completed.(Please see the revised version.).

Comment 2: Figure 7 is not clear.

Response: Thank you for your valuable advice. Figure 7 was regenerated in the data processing software with enlarged text and numeric fonts. (Please see the revised version.).

Comment 3: Please state the country of the Technical specification of quality evaluation for fertilizing machinery(NY/T 1003-2006).

Response: Thank you for your valuable suggestion.Technical specification of quality evaluation for fertilizing machinery(NY/T 1003-2006) is the Chinese standard.

Comment 4: Please give explanation whey the coefficient of variation shows a trend of decreasing as the speed of the spiral fertilizer

drain increases.

Response: Thank for your valuable suggestion. When the spiral fertilizer drain speed increases, the amount of fertilizer discharged also increases, and the increase in the amount of fertilizer discharged leads to uneven discharging, thus the coefficient of variation becomes larger.

Comment 5:  In introduction , no research papers using RSM were reviewed, specially in fertilization machines.

Response:Thank for your valuable suggestion. The research of RSM in fertilization machines has been added in the introduction.(Please see the revised version.).

Comment 6:  Provide φ (the filling factor of the spiral fertilizer drain).

Response:Thank for your valuable suggestion. φ is 0.25(the filling factor of the spiral fertilizer drain).

Comment 7: Does the fertilizer moisture content and its diameter had effect on coefficient of variation ? please give data or refences.

Response:Thank for your valuable suggestion.The moisture content of fertilizer has an effect on the coefficient of variation, for details please refer to the paper "Lu Fan. Design and experiment of cam top plate self-cleaning fertilizer apparatus[D]. Jiangxi Agricultural University", after the test, the diameter of fertilizer has no significant effect on the coefficient of variation.

---

## [Decision Letter · Decision Letter 1]

18 Oct 2022

PONE-D-22-07555R1Design and experiments of an automatic depth-adjusting double screw trencher and fertiliserningPLOS ONE

Dear Dr. song,

Thank you for submitting your manuscript to PLOS ONE. After careful consideration, we feel that it has merit but does not fully meet PLOS ONE’s publication criteria as it currently stands. Therefore, we invite you to submit a revised version of the manuscript that addresses the points raised during the review process.

Please revise.

We look forward to receiving your revised manuscript.

Kind regards,

Academic Editor

PLOS ONE

Reviewers' comments:

Reviewer's Responses to Questions

**Comments to the Author**

1. If the authors have adequately addressed your comments raised in a previous round of review and you feel that this manuscript is now acceptable for publication, you may indicate that here to bypass the “Comments to the Author” section, enter your conflict of interest statement in the “Confidential to Editor” section, and submit your "Accept" recommendation.

Reviewer #2: (No Response)

Reviewer #3: All comments have been addressed

2. Is the manuscript technically sound, and do the data support the conclusions?

Reviewer #2: Partly

Reviewer #3: Yes

3. Has the statistical analysis been performed appropriately and rigorously? 

Reviewer #2: No

Reviewer #3: Yes

4. Have the authors made all data underlying the findings in their manuscript fully available?

Reviewer #2: Yes

Reviewer #3: Yes

5. Is the manuscript presented in an intelligible fashion and written in standard English?

Reviewer #2: Yes

Reviewer #3: Yes

6. Review Comments to the Author

Reviewer #2: Dear author,

I specified my reviews about manuscript as below;

-The different aspects of the existing machine from the literature or those made in the industry should be discussed and detailed.

-Does the existing machine have different sectoral application areas or can it be preferred for different sectoral applications in the industry? It should be discussed.

-Has a cost analysis been done in terms of economy? Details should be gave about the workpiece materials and cost information should be given or a comparison should be made with other machines in the industry related to the cost.

-It is useful to give a figure of the manufacturing processes of the machine.

-How were the datas obtained with RSM verified? How many times were the tests repeated?

-Conclusions section is too short. A general evaluation should be written and the results obtained should be detailed.

Kind regards.

Reviewer #3: Please, edit the paper according to previous comments and after minor changes I recommend the paper to be published.

7. PLOS authors have the option to publish the peer review history of their article (what does this mean?). If published, this will include your full peer review and any attached files.

Reviewer #2: No

Reviewer #3: No

---

## [Author Response · Author response to Decision Letter 1]

20 Oct 2022

Dear Editors and Reviewers

Thank you very much for your email regarding my manuscript (PONE-D-22-07555). We really thank you for giving us a valuable opportunity to improve our manuscript. According to your suggestion and referees’ comments, we have carefully revised the manuscript and addressed all the comments.

The revised version with changes in RED has been submitted electronically via the Web. Enclosed please find a letter detailing our revisions and reply to the comments. I hope the revised version is suitable for publication.

I am looking forward to hearing from you.

Best regards and wishes,

Zhiyu Song

To Reviewer #1

Comment 1: English editing.

Response: Thank you for your valuable advice.English editing has been completed.(Please see the revised version.).

Comment 2: Figure 7 is not clear.

Response: Thank you for your valuable advice. Figure 7 was regenerated in the data processing software with enlarged text and numeric fonts. (Please see the revised version.).

Comment 3: Please state the country of the Technical specification of quality evaluation for fertilizing machinery(NY/T 1003-2006).

Response: Thank you for your valuable suggestion.Technical specification of quality evaluation for fertilizing machinery(NY/T 1003-2006) is the Chinese standard.

Comment 4: Please give explanation whey the coefficient of variation shows a trend of decreasing as the speed of the spiral fertilizer

drain increases.

Response: Thank for your valuable suggestion. When the spiral fertilizer drain speed increases, the amount of fertilizer discharged also increases, and the increase in the amount of fertilizer discharged leads to uneven discharging, thus the coefficient of variation becomes larger.

Comment 5:  In introduction , no research papers using RSM were reviewed, specially in fertilization machines.

Response:Thank for your valuable suggestion. The research of RSM in fertilization machines has been added in the introduction.(Please see the revised version.).

Comment 6:  Provide φ (the filling factor of the spiral fertilizer drain).

Response:Thank for your valuable suggestion. φ is 0.25(the filling factor of the spiral fertilizer drain).

Comment 7: Does the fertilizer moisture content and its diameter had effect on coefficient of variation ? please give data or refences.

Response:Thank for your valuable suggestion.The moisture content of fertilizer has an effect on the coefficient of variation, for details please refer to the paper "Lu Fan. Design and experiment of cam top plate self-cleaning fertilizer apparatus[D]. Jiangxi Agricultural University", after the test, the diameter of fertilizer has no significant effect on the coefficient of variation.

To Reviewer #2

Comment 1: The different aspects of the existing machine from the literature or those made in the industry should be discussed and detailed.

Response:Thank for your valuable suggestion. Few domestic machines are available for fertilizer application specifically for tea plantations. Moreover, the available devices can perform a specific operation at a time such as trenching, fertilizer application, and mulching, and there is no provision for automatic depth adjustment, which leads to poor efficiency of fertilizer application, poor operation quality, and uneven fertilization . Globally, more advanced automated machinery and equipment have been developed specifically for fertilizer application in tea gardens. However, these are expensive, inflexible, unable to perform multitasking, and inconsistent with domestic tea garden planting standards. Therefore, this study proposes a novel design of an automatic depth-adjusting double-spiral trenching and fertilizer application machine, which can perform trenching, fertilizer application, and mulching operations, prevent soil accumulation, and improve fertilizer application quality and efficiency in tea gardens.(Please see the revised version.)

Comment 2: Does the existing machine have different sectoral application areas or can it be preferred for different sectoral applications in the industry? It should be discussed.

Response:Thank for your valuable suggestion.The fertilizing machine in this study is mainly used for fertilising tea plantations, but can also be used for fertilising crops with a planting row spacing greater than 1400mm and a fertiliser depth of less than 330mm.

Comment 3: Has a cost analysis been done in terms of economy? Details should be gave about the workpiece materials and cost information should be given or a comparison should be made with other machines in the industry related to the cost.

Response:Thank for your valuable suggestion.Because the spiral blade completes the cutting and slanting operation to the soil, and at the same time carries out the straight line ditching operation, it requires high strength and hardness of the material, so Cr12MoV with a thickness of 10mm is chosen as the material.The rest of the machined parts are made of stainless steel.

Comment 4: It is useful to give a figure of the manufacturing processes of the machine.

Response:Thank for your valuable suggestion.A figure of the manufacturing processes of the machine has been added, so please refer to Figure 6 of the paper.(Please see the revised version.)

Comment 5: How were the datas obtained with RSM verified? How many times were the tests repeated?

Response:Thank for your valuable suggestion.The trials were validated by field trials with three replications of each group.

Comment6: Conclusions section is too short. A general evaluation should be written and the results obtained should be detailed.

Response:Thank for your valuable suggestion.The conclusion section has been added.(Please see the revised version.)

---

## [Decision Letter · Decision Letter 2]

4 Nov 2022

Design and experiments of an automatic depth-adjusting double screw trencher and fertiliserning

PONE-D-22-07555R2

Dear Dr. song,

We’re pleased to inform you that your manuscript has been judged scientifically suitable for publication and will be formally accepted for publication once it meets all outstanding technical requirements.

Kind regards,

Academic Editor

PLOS ONE

Additional Editor Comments (optional):

Reviewers' comments:

Reviewer's Responses to Questions

**Comments to the Author**

1. If the authors have adequately addressed your comments raised in a previous round of review and you feel that this manuscript is now acceptable for publication, you may indicate that here to bypass the “Comments to the Author” section, enter your conflict of interest statement in the “Confidential to Editor” section, and submit your "Accept" recommendation.

Reviewer #2: All comments have been addressed

Reviewer #3: All comments have been addressed

2. Is the manuscript technically sound, and do the data support the conclusions?

Reviewer #2: Yes

Reviewer #3: Yes

3. Has the statistical analysis been performed appropriately and rigorously? 

Reviewer #2: Yes

Reviewer #3: Yes

4. Have the authors made all data underlying the findings in their manuscript fully available?

Reviewer #2: Yes

Reviewer #3: Yes

5. Is the manuscript presented in an intelligible fashion and written in standard English?

Reviewer #2: Yes

Reviewer #3: Yes

6. Review Comments to the Author

Reviewer #2: Dear author,

Necessary revisions were performed for manuscript. Manuscript can be accepted to publish.

Kind regards.

Reviewer #3: The authors accepted the comments, I recommend the paper to be published. Thanks.

The authors accepted the comments, I recommend the paper to be published. Thanks.

The authors accepted the comments, I recommend the paper to be published. Thanks.

7. PLOS authors have the option to publish the peer review history of their article (what does this mean?). If published, this will include your full peer review and any attached files.

Reviewer #2: No

Reviewer #3: No

---

## [Editor Report · Acceptance letter]

2 Dec 2022

PONE-D-22-07555R2 

Design and experiments of an automatic depth-adjusting double screw trencher and fertiliserning 

Dear Dr. song:

I'm pleased to inform you that your manuscript has been deemed suitable for publication in PLOS ONE. Congratulations! Your manuscript is now with our production department. 

Kind regards, 

on behalf of

Dr. Robert Jeenchen Chen 

Academic Editor

PLOS ONE